# OpenReview forum: "IncentRL: Bayesian Adaptation of Preference Gaps in Reinforcement Learning"
_ICLR.cc/2026/Conference — ICLR 2026 Conference Withdrawn Submission_

### Official Review · Reviewer_P4Mg · 2025-10-23

**Soundness:** 1
**Presentation:** 3
**Contribution:** 1
**Rating:** 2
**Confidence:** 4

**Summary:**

The paper presents a framework to augment the task rewards in reinforcement learning with an intrinsic reward based on the difference in observed and desired outcomes in the environment dynamics. The framework is general enough that leaves freedom in how the preference model is designed, but only hand-crafted preferences are used throughout the paper instead of learned ones. Additionally, the paper presents ideas to autotune the parameter $\beta$ which controls the magnitude of the intrinsic incentive compared to the task rewards. In a few small MDP examples, the presented method achieves better performance than naive RL without any intrinsic incentive.

**Strengths:**

The paper is clearly written and easy to follow, and the ideas are well communicated. Each of the components of the presented framework are described in detail and in a well-structured manner. The presented framework is general, since the KL between the desired and expeted outcomes is a general idea that allows many different variants in the way both the dynamics and preference models are learned or used.
The discussion around different ways of encoding the preference model is sound, and the links to the cognitive motivation and free-energy principle are sound.

**Weaknesses:**

I find this paper to be very poorly contextualized. There are very few citations to other work in intrinsically-motivated RL [1,2,3,4,5,6] (just to list a few), or adaptations of the free energy principle and active inference theory to the RL framework [7,8]. There is a large body of work in these fields published over the last decade which the paper ommits. The related work section is very brief, and skips progress in these directions. Not only empirical progress in the form of new intrinsically-motivated RL algorithms being recently proposed, but also discussion and other practices which make intrinsic rewards work in RL [6].

I wouldn't relate to the idea of adapting $\beta$ dynamically during training as the "central novelty" of this work in the Abstract, since the runs with adaptive $\beta$ are not even in shown in the paper, but are said to achieve a similar performance to the fixed $\beta$ ones. I find the ideas of using the KL divergence between predicted and desired outcomes in the environment to be more sound.

The authors introduce the distribution $p(o|s,a)$ right after having defined fully-observed MDPs. To be correct, you should either specify that you are predicting the transition dynamics over states $p(s'|s,a)$ (which I believe is the case because the toy MDP used, MountainCar and MiniGrid are all fully-observed MDPs) or otherwise you should introduce partially-observed MDPs (POMDPs) making an explicit separation of the state space $\mathcal{S}$ and observation space $\mathcal{O}$.

I find Section 3.3 is not needed in the main paper since the discussion on the role of $\beta$ is straightforward and can be made more concise. I have a similar opinion of Sections 4.2 (the discussion and the equation is repeated in 4.1 and 4.2) and 4.3 (discussing latent representations for encoding the preference model, but not used anywhere in the paper). Instead, the authors should allocate more space in covering related work in the area and extending their evaluation.

I wouldn't call the toy MDP presented in 5.2 a "sparse-reward" problem, since with a single (state,action) pair, the agent can experience a reward in the environment with 0.3 probability, offering enough supervision for training any vanilla RL agent to solve the task.

Crucially, I don't think the provided empirical evidence supports the claims of the paper. I think the paper is missing a much broader evaluation of the method in more complex and recent benchmarks used for exploration (e.g., a subset of tasks from MiniGrid, ProcGen, Atari, Crafter ,etc.) and importantly, comparisons with existing methods designed for improved intrinsic exploration. The paper does not cover, cite, or explicitly state the differences of their method in the related work area, neither it evaluates and compares existing methods that are similar in design, have been used in the same environments, and hence are relevant baseline comparisions.

[1] Pathak, Deepak, Dhiraj Gandhi, and Abhinav Gupta. "Self-supervised exploration via disagreement." International conference on machine learning. PMLR, 2019.

[2] Guo, Zhaohan, et al. "Byol-explore: Exploration by bootstrapped prediction." Advances in neural information processing systems 35 (2022): 31855-31870.

[3] Sekar, Ramanan, et al. "Planning to explore via self-supervised world models." International conference on machine learning. PMLR, 2020.

[4] Kapturowski, Steven, et al. "Unlocking the Power of Representations in Long-term Novelty-based Exploration." Second Agent Learning in Open-Endedness Workshop. 2024.

[5] Badia, Adrià Puigdomènech, et al. "Never give up: Learning directed exploration strategies." arXiv preprint arXiv:2002.06038 (2020).

[6] Yuan, Mingqi, et al. "Rlexplore: Accelerating research in intrinsically-motivated reinforcement learning." arXiv preprint arXiv:2405.19548 (2024).

[7] Berseth, Glen, et al. "SMiRL: Surprise minimizing RL in dynamic environments." arXiv preprint arXiv:1912.05510 71 (2019).

[8] Hugessen, Adriana, et al. "Surprise-Adaptive Intrinsic Motivation for Unsupervised Reinforcement Learning." arXiv preprint arXiv:2405.17243 (2024).

**Questions:**

How can the preferences be encoded in environments that are not fully-observed MDPs? Concretely, how can LLMs help with that? (since that is mentioned in the paper, but I don't understand how that would work).

---

### Official Review · Reviewer_FsYu · 2025-10-25

**Soundness:** 1
**Presentation:** 2
**Contribution:** 2
**Rating:** 0
**Confidence:** 5

**Summary:**

This paper proposes a method to provide intrinsic rewards to an RL agent during training, based on a KL-divergence between the predicted outcome of actions vs the preferred outcome. A Bayesian modulation of the KL-parameter is claimed but not detailed in the paper. The method is demonstrated in three toy environments, demonstrating improved learning efficiency.

**Strengths:**

- Reinforcement learning under sparse rewards is a difficult challenge.
- The proposed method is intuitively interesting.
- The paper attempts a formal analysis in addition to empirical results, which is appreciated.

**Weaknesses:**

Unfortunately, the paper has some significant flaws:

1) A Bayesian modulation of the beta-parameter is claimed, but no details are provided in the paper about the method.

2) The prediction p(o|s,a) and preference q(o|s) are not clearly defined. What is the domain of "o"? For "q", the paper describes it as "what the agent would like to happen in state s"; happen after what? Should "a" be included here?

3) Proposition 1 is a main theoretical property claimed in the paper, but no full proof is provided (only a proof sketch). I am also not sure what the authors mean by "If the external reward rext admits an optimal policy π∗", because any MDP admits an optimal policy.

4) There are a number of other methods for intrinsic rewards, but the empirical evaluation does not compare to any other prior methods. For example, it would be interesting to include DeRL (see below) as a baseline.

5) Related work: a core motivation of the paper is that prior methods require careful modulation of the hyper-parameter that combines the intrinsic and extrinsic rewards. This problem is tackled in DeRL [1] (https://arxiv.org/abs/2107.08966), which not mentioned in this paper. DeRL eliminates the hyper-parameter by decoupling policy training for intrinsic and extrinsic rewards.

The paper is also quite repetitive in some places. For example, the combined reward and return are defined repeatedly in several places. I also could not see a full specification of hyper-parameters used in the evaluated RL algorithms (reproducibility).

I think this work could have potential and I hope my comments will be useful in the next version.

**Questions:**

How does the proposed method compare to a method like DeRL?

---

### Official Review · Reviewer_wdim · 2025-10-27

**Soundness:** 1
**Presentation:** 1
**Contribution:** 1
**Rating:** 0
**Confidence:** 4

**Summary:**

The work proposes a novel reward shaping approach for reinforcement learning (RL) that computes intrinsic rewards as the negative KL difference between a target distribution of outcomes and predicted outcome distributions. The outcome distributions are pre-defined and provide additional signal towards the learning of RL agents. The approach is evaluated in a toy MDP environment for illustrative purposes, as well as the MountainCar and a MiniGrid environment as sparse-reward exploration tasks. Substantial gains in efficiency can be observed in the toy MDP, while minor benefits are observed in MountainCar and Minigrid.

**Strengths:**

The problem of efficient learning under sparse rewards is relevant and an impactful problem to tackle. The idea of incorporating additional information from preferred outcomes is conceptually interesting, albeit not clearly executed (see weaknesses below).

**Weaknesses:**

Overall, I am afraid that the work is clearly not of sufficient quality to be considered for acceptance at ICLR. Below, I try to provide key weaknesses that I believe should be addressed and would substantially strengthen the work.

## Originality and Prior Work

1. To start with, the work only loosely discusses prior work on intrinsic motivation and reward shaping in Section 2.1. There is a rich space of literature that ranges from future predictions (ICM and RND being cited), state visitation counts [4], density functions of states [3], and combinations of several schemes [1, 2] just to mention few -- all propose ways to determine "novelty" or interestingness of states for exploration. I would advice to look at the literature in this space in more detail.
2. In addition to a rich space of literature on defining intrinsic rewards for sample efficient learning, there also exists prior literature on balancing intrinsic and extrinsic rewards, similar to the proposed Bayesian approach of adapting $\beta$. Some examples are [5, 6]
3. The work continually makes connections to the free energy principle and dopamine frameworks, but these are merely described as loose connections. It would be helpful if the authors would provide citations, definitions, and clearly outline any connections that they believe add to their work.

[1] Raileanu, Roberta, and Tim Rocktäschel. "Ride: Rewarding impact-driven exploration for procedurally-generated environments." _arXiv preprint arXiv:2002.12292_ (2020).

[2] Zhang, Tianjun, Huazhe Xu, Xiaolong Wang, Yi Wu, Kurt Keutzer, Joseph E. Gonzalez, and Yuandong Tian. "Noveld: A simple yet effective exploration criterion." _Advances in Neural Information Processing Systems_ 34 (2021): 25217-25230.

[3] Bellemare, Marc, Sriram Srinivasan, Georg Ostrovski, Tom Schaul, David Saxton, and Remi Munos. "Unifying count-based exploration and intrinsic motivation." _Advances in neural information processing systems_ 29 (2016).

[4] Tang, Haoran, Rein Houthooft, Davis Foote, Adam Stooke, OpenAI Xi Chen, Yan Duan, John Schulman, Filip DeTurck, and Pieter Abbeel. "# exploration: A study of count-based exploration for deep reinforcement learning." _Advances in neural information processing systems_ 30 (2017).

[5] Schäfer, Lukas, Filippos Christianos, Josiah P. Hanna, and Stefano V. Albrecht. "Decoupled reinforcement learning to stabilise intrinsically-motivated exploration." _Autonomous agents and multi-agent systems (AAMAS) conference_ (2022).

[6] Chen, Eric, Zhang-Wei Hong, Joni Pajarinen and Pulkit Agrawal. “Redeeming Intrinsic Rewards via Constrained Optimization.” _Advances in neural information processing systems_ (2022).

## Clarity of Methodology and Experiments
Below, I list further concerns I have regarding a lack of clarity in the defined method and experiments conducted as part of this work.
### Methodology
4. The introduction and Section 3 that defined the proposed IncentRL approach, makes use of "outcomes" in the form of predicted and preferred outcome distributions. However, it is never defined and not quite clear to me what these outcomes are. Are outcomes future states that might be predicted, or some specific quantity of states?
	1. Related to what outcomes are, it is not clear to me where the outcome distributions are coming from either and what assumptions are being made. It appears that the work assumes that preferred outcome distribution are specified to determine a preference over outcomes. But how do you specify the predicted outcome distribution $p(o | s, a)$? Section 3.1 mentions that this quality "[...] may be obtained from a forward model or from environment dynamics" but it is unclear to me which of these is being done throughout experiments. Do you train a forward model from experience tuples or do you assume access to the environment transition function?
5. In Section 5.4 of the experimental section, a Bayesian adaptation of $\beta$ is being evaluated, however, none such adaptation scheme of $\beta$ is described when the method is introduced in Section 3?
6. The theoretical contributions in proposition 1 and 2 both lack formal proofs. I would expect these to be at least provided as part of the Appendix. Also, both of these propositions do not appear to be particularly insightful since they merely talk about the extreme cases of extremely small and large $\beta$ values to draw fairly obvious conclusions.
### Experiments
7. In Section 5.1, the "Algorithm" paragraph mentions that the agent estimates its belief about possible next outcomes, but it is never defined how these predictions are obtained (connected to weakness 4.1).
8. The predicted and preferred outcome distributions for most experiments are incompletely defined or entirely undefined. Given these are central part of the main contribution of this work, this appears to be a lack of critical details.
	1. For Experiment 1 (Section 5.2) only defines predicted and preferred outcome distributions for the 2-state MDP for state $s_0$. The rest of the distributions appear undefined.
	2. For Experiment 2 (Section 5.3) in MountainCar, the distribution is described as "preferred outcome assigning all probability to the goal ($x \geq 0.5$), and $p(o | s, a)$ is the predicted outcome distribution" but those are still vague to me. The described preferred outcome distribution appears to now be defined as a property of states $x \geq 0.5$ rather than full states as before (connected to problem of lack of definition of what outcomes are, see weakness 4), and the predicted outcome distribution is fully undefined.
	3. For Experiment 3 (Section 5.4), these quantities appear entirely undefined.
9. According to the description of training details in Section 5.4 on Experiment 3, the results are averaged over 3 independent seeds but Figure 2 visualizes no indication of dispersion/ deviation. I would expect a visualisation of the mean and shading to indicate dispersion (e.g. standard deviation, standard error, min/ max). Similarly, the caption of Table 2 states "$\pm$ std over 3 seeds" but then no such deviations are shown in the table.
10. It appears that in experiment 2 and 3, IncentRL provides no clear benefits. According to Table 2, only for one value of $\beta$ did IncentRL slightly outperform the base algorithm ($\beta = 0$) and it is not clear that these gains are significant without indication of deviation. For other values of $\beta$, performance dropped significantly with IncentRL compared to the baseline. In Figure 2, results also don't seem to show a very significant change from the base algorithm to IncentRL.
11. Figure 3 visualizes the (undefined; see weakness 5) Bayesian adaptation of $\beta$ and claims that "the posterior mean quickly concentrates near the effective region ($\beta \sim 0.1$)" but the Figure more so suggests that the posterior mean moves away from $0.1$ over several rounds and might converge closer to $0.02$ which I would not consider near the effective region of $0.1$.

## Significance
12. From a high-level perspective, this work states that, when given preferred outcomes, the algorithm can leverage this information under (somewhat unclear) assumptions to shape rewards and then learn more efficiently. This does not appear like a very novel or significant contribution but represent a typical instance of reward shaping. This puts into question whether this work makes significant contributions that warrant publication.

**Questions:**

1. How would you define "outcomes" that the predicted and preferred outcome distributions are defined over? (Weakness 4 for more details)
2. How do you obtain the predicted outcome distributions? Are these assumed to be given (like the preferred outcome distributions) or learned? (Weakness 4.1 and 7 for more details)
3. Could you describe the Bayesian adaptation process of $\beta$ that is being evaluated in Section 5.4? (Weakness 5 for more details)
4. How are the predicted and preferred outcome distributions defined for each of the experiments? (Weakness 8 for more details)
5. Experiments are stated to be repeated for 3 independent seeds but no metric of dispersion is being provided in Table 2 or Figure 2. Would the authors be able to provide standard deviation or any other indication of dispersion for these experiments to judge the significance of provided results?

---

### Official Review · Reviewer_yftj · 2025-10-31

**Soundness:** 1
**Presentation:** 1
**Contribution:** 1
**Rating:** 0
**Confidence:** 5

**Summary:**

This paper studies model-free RL and proposes to add a particular type of intrinsic reward to the standard extrinsic reward in order to boost performance. Specifically, they propose to be given a distribution over preferred outcomes conditioned on state and then to subtract the KL-divergence between this preferred outcome distribution and a predicted outcome distribution. Experiments on toy RL environments show that the approach potentially leads to more data efficient learning.

**Strengths:**

- The method proposed is novel to the best of my knowledge.
- I really like how the authors are connecting to other frameworks for intelligent behavior such as the free energy principle and ideas in cognitive science. I think the direction is interesting, despite the concerns I raise below about this particular instantiation of writing up the work.

**Weaknesses:**

- The clarity of the paper could be significantly improved. For example, it would be helpful to specify formally what outcomes are. I wasn't sure if they were resulting next states or something else. A Bayesian adaptation scheme is mentioned multiple times (including the abstract), but never defined.
- In several places it seems like there are bullet points embedded in paragraphs, suggesting that the paper was written last minute and still needs careful editing.
- Motivation: it is unclear what the preferred outcome distribution is (formally) and why it is reasonable to expect a learning agent to have such a distribution.
- Main method confusion: it was not clear if and how the agent's predicted outcome distribution was updated during learning.
- In the empirical study, a small number of trials are reported and confidence intervals are wide (but type unspecified). Please see "Empirical Design in Reinforcement Learning" for great discussion on why these details matter in empirical RL research.

**Questions:**

Discussion of the weaknesses given above would be the most productive use of the rebuttal.

---

### Note · Authors · 2025-11-12

**Comment:**

We thank the reviewers and area chair for their feedback. We have decided to withdraw this submission and will prepare a revised version for a future venue.

**Withdrawal Confirmation:**

I have read and agree with the venue's withdrawal policy on behalf of myself and my co-authors.